# Synergistic Effect between the *APOE ε4* Allele with Genetic Variants of *GSK3B* and *MAPT*: Differential Profile between Refractory Epilepsy and Alzheimer Disease

**DOI:** 10.3390/ijms251810228

**Published:** 2024-09-23

**Authors:** Danira Toral-Rios, Pavel Pichardo-Rojas, Elizabeth Ruiz-Sánchez, Óscar Rosas-Carrasco, Rosa Carvajal-García, Dey Carol Gálvez-Coutiño, Nancy Lucero Martínez-Rodríguez, Ana Daniela Rubio-Chávez, Myr Alcántara-Flores, Arely López-Ramírez, Alma Rosa Martínez-Rosas, Ángel Alberto Ruiz-Chow, Mario Alonso-Vanegas, Victoria Campos-Peña

**Affiliations:** 1Department of Psychiatry, School of Medicine, Washington University, St. Louis, MO 63110, USA; rios@wustl.edu; 2The Vivian L. Smith Department of Neurosurgery, McGovern Medical School, The University of Texas Health Science Center at Houston, Houston, TX 77030, USA; pavel.s.pichardorojas@uth.tmc.edu; 3Neurochemistry Laboratory, National Institute of Neurology and Neurosurgery “Manuel Velasco Suárez”, Mexico City 14269, Mexico; ruizruse@yahoo.com.mx; 4Geriatric Assessment Center, Department of Health, Iberoamerican University, Mexico City 01219, Mexico; oscar_rosas_c@hotmail.com; 5SINANK’AY Geriatric Center, Jurica, Santiago de Querétaro 76100, Mexico; rosacarvajal@sinankay.net; 6Experimental Laboratory of Neurodegenerative Diseases, National Institute of Neurology and Neurosurgery “Manuel Velasco Suárez”, Mexico City 14269, Mexico; galvezcoutinho@gmail.com (D.C.G.-C.); arelymlopezr@gmail.com (A.L.-R.); 7Epidemiological Research Unit in Endocrinology and Nutrition, Children’s Hospital of Mexico Federico Gómez, Mexico City 06720, Mexico; amr70@hotmail.com; 8High Specialty Medical Unit (UMAE), Specialty Hospital, National Medical Center (CMN), XXI Century, Mexico City 06720, Mexico; anadanielarubio@gmail.com; 9Department of Psychiatry, National Institute of Neurology and Neurosurgery “Manuel Velasco Suárez”, Mexico City 14269, Mexico; fioremyr@gmail.com (M.A.-F.); angel.ruiz@innn.edu.mx (Á.A.R.-C.); 10Cognition and Behavior Unit, National Institute of Neurology and Neurosurgery “Manuel Velasco Suárez”, Mexico City 14269, Mexico; appliedneuropsychology@gmail.com; 11Director of the International Center for Epilepsy Surgery, HMG-Coyoacan Hospital, Mexico City 04380, Mexico; alonsovanegasm@gmail.com

**Keywords:** Alzheimer disease, epilepsy, SNP, drug-resistant Epilepsy, *APOE ε4* allele, hippocampal sclerosis

## Abstract

Temporal Lobe Epilepsy (TLE) is a chronic neurological disorder characterized by recurrent focal seizures originating in the temporal lobe. Despite the variety of antiseizure drugs currently available to treat TLE, about 30% of cases continue to have seizures. The etiology of TLE is complex and multifactorial. Increasing evidence indicates that Alzheimer’s disease (AD) and drug-resistant TLE present common pathological features that may induce hyperexcitability, especially aberrant hyperphosphorylation of tau protein. Genetic polymorphic variants located in genes of the microtubule-associated protein tau (*MAPT*) and glycogen synthase kinase-3β (*GSK3B*) have been associated with the risk of developing AD. The *APOE ε4* allele is a major genetic risk factor for AD. Likewise, a gene-dose-dependent effect of *ε4* seems to influence TLE. The present study aimed to investigate whether the *APOE ɛ4* allele and genetic variants located in the *MAPT* and *GSK3B* genes are associated with the risk of developing AD and drug-resistant TLE in a cohort of the Mexican population. A significant association with the *APOE ε4* allele was observed in patients with AD and TLE. Additional genetic interactions were identified between this allele and variants of the *MAPT* and *GSK3B* genes.

## 1. Introduction

Epilepsy is one of the most common chronic neurological diseases, characterized by the appearance of recurrent unprovoked seizures [1]. Epilepsy affects around 50 million people worldwide [2]. Seizures in most epilepsy cases originate in the temporal lobe, and about 30% of patients with temporal lobe epilepsy (TLE) may develop drug resistance [3]. The etiology of TLE is complex and multifactorial; however, the most important causes are hippocampal sclerosis (HS), tumors, traumatic brain injury, and infections [4]. The vast majority of TLE cases course to HS, a neurodegenerative process in mesial regions, such as the hippocampus, amygdala, and entorhinal cortex, which are also commonly affected in patients with Alzheimer’s disease (AD) [5,6,7,8]. Mesial temporal lobe epilepsy (MTLE) patients develop a progressive cognitive decline [9,10,11,12]. Additional evidence suggests an association between AD and TLE that may reveal new mechanisms and pharmacological targets to treat both conditions. Epidemiological studies have shown an increased prevalence of AD and other dementias, with epileptic seizures as a comorbidity [13,14,15,16]. Neuropathological extracellular aggregation of amyloid-ε plaques and intracellular accumulation of neurofibrillary tangles (NTF) composed of the microtubule-associated mainly hyperphosphorylated protein tau have been found in the brain tissue of patients with refractory TLE [17,18]. The role of tau protein in modulating neuronal hyperexcitability has been shown in mouse models of AD and in chemically-induced seizure models, where tau-reducing therapy decreased the severity of seizures and improved cognitive impairments [19]. The underlying mechanisms of tau hyperphosphorylation in epilepsy involve alteration of tau kinases like glycogen synthase kinase-3β (GSK-3β) and cyclin-dependent kinase 5 (CDK5); overactivation of both has been detected in resected tissue from refractory patients with epilepsy [20]. Hyperexcitability mediated by tau in patients with refractory epilepsy might be influenced by APOE, the major genetic risk factor for AD. Thus, neurons in patients with TLE carrying the *APOE ε4* allele are less resistant to damaging hyperexcitability associated with epilepsy than *APOE ε3* carriers [21]. In the present study, we aimed to determine whether the presence of single nucleotide polymorphisms (SNPs) in genes encoding the microtubule-associated protein tau (*MAPT*), the kinase *GSK3B*, and two heat shock proteins involved in tau aggregation (HSPs) are associated with the risk of developing AD and drug-resistant TLE in a cohort of the Mexican population. In addition, the genetic susceptibility of the *APOE ε4* allele, as well as the genetic interaction with the abovementioned SNPs, were analyzed.

## 2. Results

### 2.1. Study Population

A total of 649 subjects were enrolled in this study: 100 patients with LOAD, 106 LOAD age-matched healthy controls (>60 years), 198 patients with refractory temporal lobe epilepsy (49 subjects with nHS-TLE, 79 with HS-TLE, and 70 with TA-TLE), and 245 epilepsy age-matched healthy controls. The population demographics (age and sex) are summarized in Table 1. No significant differences in sex (nHS-TLE *p* = 0.52, HS-TLE *p* = 0.08, TA-TLE *p* = 0.16, and AD *p* = 0.47) or age (nHS-TLE *p* = 0.101, HS-TL *p* = 0.011, TA-TLE *p* = 0.167, and AD *p* = 0.178) only for the group HS-TLE *p* = 0.011, distribution was observed among the study groups and respective age-matched controls.

### 2.2. Allelic and Genotypic Distribution

The allele and genotype frequencies of the polymorphic variants located in the genes *MAPT*, *HSPA1L, HSPA5, GSK3B*, and APOE are summarized in Table 2. Genotype distributions of SNPs located in HSPA1L and HSPA5 (rs2227956 and rs391957) were not in Hardy–Weinberg equilibrium (*p* < 1  × 10^−3^) and were consequently excluded from further analysis (Appendix A). Our results did not show significant differences in allele or genotype frequencies of the SNPs located in the *MAPT* and *GSK3B* genes between LOAD and healthy aged-matched controls or patients with TLE and respective controls. *APOE ε4* carrier status showed a significant association with AD risk in our cohort (*p* = 0.0001, OR > 2) (Table 2).

A significant association with the *APOE ε4* allele was observed in patients with TA-TLE (*p* = 0.038) and patients with HS-TLE (*p* = 0.024). Our results indicate that the polymorphic variants studied in the MAPT and *GSK3B* genes are not associated with the risk of developing LOAD and nHS-TLE in the samples analyzed, whereas allele *ε4* shows a strong association with both neurological conditions.

### 2.3. Genetic Inheritance Models and ApoE ε4 Interaction

Disease susceptibility of SNPs in the *MAPT* and *GSK3B* genes under specific genetic inheritance models was analyzed (Table 3). No significant association was observed between the SNPs on that gene and susceptibility to developing AD and nHS-TLE. However, after performing multinomial regression to consider their association with the *APOE ε4* allele, significant associations were detected, suggesting a potential genetic interaction effect. The majority of these differences were present in patients with nHS-TLE and AD. For the nHS-TLE group, a significant association with the e4 allele was identified with variants rs242557 (*p* = 0.038, OR = 11.366, 95%CI (1.147–112.648)) and rs1467967 (*p* = 0.003, OR = 47.717, 95%CI (3.727–610.877)) located in the *MAPT* gene using a recessive model, and the variant rs6438552 (OR = 13.523, 95%CI (2.265–80.736), *p* = 0.004) located in the *GSK3B* gene ( using the heterozygote model). In patients with AD, *APOE ε4* carrier status showed a significant correlation with the genetic variant rs242557 (*p* = 0.029, OR = 4.862, 95%CI (1.171–20.187)) and rs2471738 (*p* = 0.043, OR = 6.4, 95%CI (1.063–38.798)) located in the *MAPT* gene, both assuming a heterozygote model. No significant correlation was observed between patients with TA-TLE and patients with HS-TLE.

### 2.4. Linkage Disequilibrium Analysis

Linkage disequilibrium analysis LD is relevant for detecting associations between genetic variants located in nearby sites [22]. No LD was detected between the SNPs located in the *MAPT* gene (Appendix A). A significant LD was observed between the variants rs334558 and rs6438552 located in the *GSK3B* gene, which allowed us to identify a haplotype block (Figure 1) composed of four unique allele combinations (AA, GG, GA, and AG). The frequency of the GA haplotype was significantly elevated in patients with HS-TLE (*p* = 0.000, OR = 3.71, 95%CI (1.70–8.12)), patients with nHS-TLE (*p* = 0.000, OR = 5.84, 95%CI (2.60–13.13)), and TA-TLE patients compared to age-matched controls (*p* = 0.000, OR = 3.88, 95%CI (1.75–8.6)). The haplotype AG was found more frequently in patients with HS-TLE (*p* = 0.000, OR = 10.34, 95%CI (3.55–30.1)), patients with nHS-TLE (*p* = 0.000, OR = 9.59, 95%CI (2.96–31.10)), and TA-TLE patients (*p* = 0.000, OR = 11.56, 95%CI (4.0–33.45)) than in the controls. No significant differences in haplotype frequencies were observed in patients with AD, suggesting that haplotypes in the *GSK3B* gene may only account for the risk of developing refractory temporal epilepsy.

### 2.5. Evaluation of Gene–Gene Interactions by Multifactorial Dimensionality Reduction

We previously identified the genetic association of the *APOE ε4* allele with genetic variants located in the *MAPT* and *GSK3B* genes in patients with AD and TLE (with and without HS) by multinomial regression, a method that may increase the probability of false positives. Gene−gene interaction was assessed with Multifactorial Dimensionality Reduction (Table 4), an analysis method that reduces the dimensionality of multilocus data to improve the ability to detect genetic combinations that confer disease risk [23]. Using MDR, a genetic interaction model in patients with nHS-TLE was identified and represented with a dendrogram and circle graph. The model includes SNPs previously associated with nHS-TLE patients, two polymorphisms in *MAPT* (rs242557/A allele and rs1467967/G allele), one polymorphism in *GSK3B* (rs6438552/G allele), and the presence of the *APOE ε4* allele. The combination of these possible interactive polymorphisms in the model yielded a maximum CVC of 10/10 and a maximum testing accuracy of 0.632 (*p* < 0.001, OR = 6.291, 95%CI (2.831–13.97)). The dendrogram (Figure 2a) and circle plot (Figure 2b) suggest a strong genetic interaction between the variant rs1467967 in the *MAPT* gene and the *APOE ε4* allele (red line, 2.74%). Certainly, both can also interact with the variant rs6438552 in the *GSK3B* gene (red line, 2.58%). However, the variant rs242557 was located on another remote branch, indicating that it may have less or a weaker relationship with other SNPs (Figure 2b, orange lines).

In the group of patients with HS-TLE, it was only possible to observe a strong interaction between allele A of the rs7521 polymorphism of *MAPT* and *APOE ε4*. MDR analysis (Figure 2c,d) showed a significant (*p* < 0.007; OR = 2.108 95%CI (1.222–3.635)) genetic interaction between the *APOE ε4* allele and the A allele of the genetic variant rs7521 located in the *MAPT* gene (red line). The combination of these polymorphisms yielded a maximum CVC of 10/10, maximum testing accuracy of 0.588 (Table 4), and positive entropy of 2.53% (Figure 2d).

In patients with AD, an interaction model was observed, including three polymorphisms present in the *MAPT* gene rs242557 (A allele), rs2471738 (T allele), and rs3785883 (A allele) with the *APOE ε4* allele (*p* < 0.001; OR = 5.221 95%CI (2.697–10.108)) (Table 4). The dendrogram (Figure 2e) and circle graph (Figure 2f) suggest a strong genetic interaction between the variant rs3785883 in the *MAPT* gene and the *APOE ε4* allele (red line, 2.47%); both can interact with the variant rs242557 (red line, 2.01%). However, the variant rs242557 was located on a remote branch, indicating a weak interaction between *APOE ε4* and other *MAPT* variants (orange lines). No linkage disequilibrium was detected in the SNPs located in the *MAPT* gene. In summary, the MDR resulted in a sensitive method for detecting multiloci (*APOE ε4*, *MAPT*, and *GSK3B*) genetic interactions. A unique genetic interaction between *MAPT*, *GSK3B*, and *APOE ε4* was identified in patients with nHS-TLE, which differs from the genetic interaction between *MAPT* and *APOE ε4* in patients with HS-TLE and patients with AD.

### 2.6. Cognitive Function in Patients with Epilepsy

A total of 42 patients with TLE, 23 with HS-TLE (29% of the total number of patients with HS) and 19 with nHS-TLE (39% of the total number of patients with nHS-TLE), were subjected to a neuropsychological battery to assess their cognitive functions after epilepsy resective surgery (Appendix A). Neuropsychological tests were grouped into nine domains (Appendix A), patients were categorized according to their *APOE ε4* carrier status, and their cognitive impairment outcome was classified into severe or non-severe impairments (Appendix A). Neuropsychological assessment in patients with HS-TLE (Appendix A) and patients with HS-TLE (Appendix A) showed a similar proportion of *APOE ε4* carriers and non-carriers presenting severe and non-severe cognitive impairments. Hence, *APOE* alleles do not seem to influence neuropsychological outcomes and cognitive functions in refractory epilepsy patients in our cohort.

## 3. Discussion

About 30% of epilepsy cases may develop resistance to antiseizure drug therapies [3]. In cases of focal temporal lobe epilepsy, resective neurosurgery is a safe and effective treatment despite the risk of complications and impacts on cognition and neuropsychiatry changes [24]. Understanding the molecular mechanisms involved in drug resistance to TLE is important for developing accurate treatments. Patients with TLE share pathological hallmarks with patients with AD, such as aggregation of amyloid beta peptide and tau protein [17,18]. Genetic association studies in patients with AD have revealed key risk factors involved in amyloid and tau aggregation pathogenesis. The major genetic risk factor for late-onset AD is the *ε4* allele of Apolipoprotein E (*APOE ε4*). Similarly, it has recently been proposed that the presence of the *APOE ε4* allele is associated with the development of refractory epilepsy [25,26]. The presence of the *ε4* allele was associated with a significant increase in tonic-clonic seizures induced by the injection of Pentylenetetrazole in a mouse model with and without AD familial mutations [27]. *APOE ε4* allele involves mechanisms associated mainly with amyloid-β aggregation that are considered fundamental for tau aggregation [28]. In multiple ethnic groups, the gene dose effect of the *APOE ε4* allele varies. The presence of a copy of the *APOE ε4* allele increases AD risk in Caucasians by 3-fold, whereas two copies increase the risk by 12-fold relative to the *ε3* allele [29]. However, the frequency of the allele *ε4* is lower, and either one or two copies of this allele have been found to increase AD risk by 2-fold [29]. Correlating with this study and a previous report conducted in a Mexican population cohort of patients with AD [30], we observed a higher frequency of Apoε3 carriers and low frequency in ε2 and ε4 carriers. In addition, our results showed a significant association between the presence of the *APOE ε4* allele and the development of AD, TA-TLE, and HS-TLE in our Mexican population cohort. The *APOE ε4* allele has been associated with an increased risk of drug-resistant epilepsy [26,31,32], but other studies have failed to prove this relationship in patients with TLE from Italy [33] or patients with TLE from Turkey without HS [34]. Discrepant results are very common in genetic association studies and may be related to the genetic background of each population, sample size effect, and interaction of several genes and environmental factors [35].

In the present work, we assessed whether variants in genes involved in tau pathogenesis would confer a risk of developing AD or drug-resistant TLE (nHS-TLE, HS-TLE, and TA-TLE) independently or by assuming a genetic interaction with the presence of the *APOE ε4*. Previous genetic studies in Caucasic and Asian populations have observed an association between polymorphic variants in microtubule-associated protein tau (*MAPT*) and AD genetic risk [36,37,38,39], while other authors have reported no association [40,41,42]. None of the variants located in the *MAPT* gene analyzed in the current work showed an association with the risk of developing AD and refractory epilepsy in our population. A common inversion divides the *MAPT* gene into two major haplotypes, H1 and H2. The H1 haplotype may affect gene expression and increase *MAPT* transcript levels [37,42,43]. Despite the H1 haplotype being present at a higher frequency relative to the H2 haplotype frequency in our study population, no association with AD and drug-resistance epilepsy risk was observed. Our data do not correlate with the H1 haplotype association with AD found in samples of a US population [43] but is consistent with what has been observed in a cohort of the Chinese population [44].

Hyperphosphorylation of tau protein decreases the binding affinity and stability of microtubules and affects the cytoskeleton dynamics of neurons. One of the main tau kinases that is key to neurofibrillary tangle formation is glycogen synthase kinase-3 (GSK3β) [45]. Overexpression of GSK3β in forebrain neurons in mice leads to tau hyperphosphorylation, neuronal death, reactive astrocytosis, and microgliosis [46,47]. Polymorphisms in the *GSK3B* locus have been previously reported to be associated with AD [37] and mood disorders [48,49,50]. No association between this variant and AD or refractory epilepsy was identified in our samples.

The pathological hyperphosphorylation of tau and inefficient removal of tau aggregates by heat shock proteins (HSPs) leads to tau dysfunction, causing the formation of NFTs and neurodegeneration. We analyzed the distribution of the allelic and genotypic frequencies of polymorphisms in the *HSP70-1* and *HSP70-5* genes. Although both SNPs presented a Hardy–Weinberg equilibrium deviation, the results showed no association with the disease in Mexican patients with AD. Similar results have been reported for Caucasian and Asian cohorts [51].

We next investigated the impact of gene–gene interactions on AD and refractory epilepsy risk. Interaction analysis between the *APOE ε4* allele and the genetic variants in *MAPT* and *GSK3B* revealed specific allelic combinations for each pathology. Thus, a genetic interaction was identified between the *APOE ε4* allele and three polymorphisms located in the *MAPT* gene (rs242557, rs2471738, and rs3785883) in patients with AD. A significant interaction between the *APOE ε4* allele and one variant in the *MAPT* locus (rs7521) was identified in refractory epilepsy patients with HS. Unlike these pathologies, synergy between the *APOE ε4* allele and genetic variants of the *MAPT* and *GSK3B* genes was found to be associated with the risk of developing nHS-TLE. No *APOE ε4* allele interaction effect was found in patients with TA-TLE, indicating that mechanisms involving tau phosphorylation are not relevant in tumor-associated epilepsy.

Complex diseases, such as AD and refractory epilepsy, are presumed to be the result of interactions between many genes (epistasis) and environmental factors [52]. Gene−gene interaction analysis can reveal gene regulatory networks and identify biochemical pathways involved in conferring susceptibility to complex diseases [53]. Our results suggest that the interaction of *APOE ε4* and tau converges in AD and HS-TLE; both diseases present striking neurodegeneration that may be strongly mediated by mechanisms involving APOE and Tau.

The *APOE ε4* (APOE4) isoform may exacerbate neuronal death through mechanisms involving tau phosphorylation. Expression of the APOE4 isoform promotes tau-dependent neuroinflammation and neurodegeneration in mice expressing human P301S mutant tau [54,55]. In addition, our results indicate that the presence of the *APOE ε4* allele in combination with *GSK3B* and *MAPT* variants can confer an increased risk of developing nHS-TLE. In this sense, the APOE4 isoform can increase GSK3β activity, inhibiting the WNT signaling pathway through its interaction with LRP5/6 receptors, leading to tau hyperphosphorylation [56].

Future functional genomic and proteomic studies must be conducted to validate the epistasis effects observed in our current work. Moreover, it would be interesting to identify whether tau aggregates differ between AD and refractory epilepsy tissues.

Association of the *APOE ε4* allele with cognitive decline in AD has been reported [57] with some discrepancies due to the diversity of methods utilized for assessing cognition and intrinsic genetic and demographic variability in the analyzed populations [58]. A previous study in patients with drug-resistant TLE before undergoing temporal lobectomy revealed an association between the *APOE ε4* allele and memory performance using the Wechsler Memory Scale. Interestingly, surgery did not improve memory performance in *APOE ε4* carriers [59]. In the present study, nHS-TLE and HS-TLE were assessed using a neuropsychological battery. We did not observe a significant difference in the performance of cognitive domains between *APOE ε4* carriers and non-carriers. One possible explanation for our lack of association is the small sample size of the assessed patients. In addition, a different battery of memory tests should be conducted in future studies.

The identification of an interaction between the *ε4* allele of *APOE* and the genetic variants of *GSK3B* and *MAPT* is one of the main strengths of this study. This interaction also allowed us to observe a specific genetic profile between the different epilepsy groups and patients with AD. This finding may, in the future, open new lines of research that will help identify specific therapeutics for each pathology. At the same time, it contributes to the study of a population with low representativeness in analyzing possible genetic factors associated with the risk of developing epilepsy and AD, such as the Mexican population. The main limitation of this work was the sample size, as well as the extension of the analysis of cognitive tests, which allowed the establishment of a close correlation between the findings and cognitive impairment.

## 4. Materials and Methods

### 4.1. Subjects

We performed a genetic case-control study on unrelated individuals from Mexico City who knew how to read and write, provided informed consent to participate in this study, and whose families had been born and lived in Mexico for at least three generations. The full population demographics are summarized in Table 1. The study was conducted following the ethical standards of the Committee on Human Experimentation of the INNN (protocols No. 100/07 and No. 45/16), according to the Declaration of Helsinki.

(a)Patients: Patients with late-onset Alzheimer’s disease (LOAD) and temporal lobe epilepsy (TLE) were analyzed in the present genetic study. Patients with late-onset Alzheimer’s disease (age onset ≥ 60 years, *n* = 100) were recruited from the General Hospital of Mexico (HGM), Hospital Angeles Mocel, and the National Institute of Neurology and Neurosurgery of Mexico (INNN), all of which are located in Mexico City. The inclusion criteria were as follows: (1) individuals over 60 years of age and (2) who had been clinically diagnosed with AD by a group of experts, using the criteria of the National Institute of Neurological and Communicative Diseases and Stroke/Alzheimer’s Disease and Related Disorders Association (NINCDS-ADRDA) [60]. Subjects with any other type of neurodegenerative or neuroinfectious disease and patients with neoplasia were excluded from this study. Patients clinically diagnosed with refractory temporal lobe epilepsy (18–60 years, *n* = 198) who underwent epilepsy resective surgery were recruited from the Epilepsy Priority Program ‘PPE’ of the INNN and the HGM hospital. The inclusion criteria for clinically diagnosed individuals with refractory epilepsy were as follows: (1) having received at least two drug regimens at appropriate therapeutic doses for at least 6 months, (2) with neurological monitoring, and (3) having seizures at a frequency of at least 3 per month [61]. TLE cases were stratified according to their pathological findings into patients with Hippocampal Sclerosis-TLE (HS-TLE, *n* = 79), patients with non-Hippocampal Sclerosis-TLE (nHS-TLE, *n* = 49), and patients with Tumor-Associated-TLE (TA-TLE, *n* = 70). Subjects with any other type of neurodegenerative disease were excluded from the study.(b)Controls. Healthy individuals were recruited as controls and stratified by age into two groups as follows: Controls for patients with LOAD and Controls for patients with TLE. Controls for the patients with LOAD (*n* = 106) were recruited at the INNN and HGM hospitals following the following inclusion criteria: (1) subjects older than 60 years and (2) clinically diagnosed as non-demented according to the NINCDS-ADRD criteria. Subjects diagnosed with any neurodegenerative disease or with a family genetic history of neurodegenerative disease were excluded from the study. Controls for patients with TLE (*n* = 245) were recruited at the INNN and HGM hospitals using the following inclusion criteria: (1) subjects around 18 and 60 years old (2) clinically diagnosed as healthy. Subjects diagnosed with any neurological disease or with a family genetic history of neurological disease were excluded from the study.

### 4.2. DNA Extraction and SNP Genotyping

DNA was isolated from peripheral blood using the QIAamp DNA Blood Midi Kit (Qiagen, Valencia, CA, USA) and stored at −80 °C until use. Genotyping of eleven single nucleotide polymorphisms (SNPs) was performed using TaqMan SNP assays (Appendix A) and reagents (Applied Biosystems, Foster City, CA, USA) in the ABI FAST 7500 thermocycler (Applied Biosystems, CA). Five SNPs (rs242557, rs1467967, rs2471738, rs7521, and rs3785883) were located in the microtubule-associated protein tau (*MAPT*) gene, one (rs2227956) in the heat shock protein family A (Hsp70) member 1 like (*HSPA1L*) gene, one (rs391957) in the heat shock protein family A (Hsp70) member 5 (*HSPA5*) gene, and two (rs334558 and rs6438552) in the gene of the glycogen synthase kinase 3 beta (*GSK3B*). Apolipoprotein E (*APOE*) alleles were determined with the variants rs429358 and rs7412, which define the *APOE ε4* and *ε2* alleles, respectively, as previously detailed [62].

### 4.3. Statistical Analyses

Allele and genotype frequencies and the Hardy–Weinberg equilibrium (HWE) were calculated with Bińkowski J, Miks S. Gene-Calc [Computer software], 2018, avalaible in www.gene-calc.pl. accessed on 2 October 2023. Genetic association study of SNPs was performed using the Statistical Package for the Social Sciences (IBM^®^SPSS Statistics 20, IBM, Armonk, NY, USA) (SPSS).

For each polymorphic variant, an association under three genetic models [63] (dominant, recessive, and heterozygote) was assessed using binary logistic regression with adjustment for age and sex in SPSS (IBM^®^SPSS Statistics 20, IBM, Armonk, NY, USA) [64]. The model with the highest likelihood was considered to be the best-fitting genetic model for each SNP. Further, a multinomial regression model was used to determine the association between the *APOE ε4* carrier status and SNP frequencies under the three different genetic models, adjusting for age and sex as potential confounders.

Linkage disequilibrium (LD) and haplotypic frequencies were determined using Haploview 4.2 (Whitehead Institute for Biomedical Research, Cambridge, MA, USA).

### 4.4. Multifactorial Dimensionality Reduction (MDR)

The MDR method was used to assess epistasis and generate an optimal one-dimensional model to predict disease susceptibility using the MDR v3.0.2 software package available at www.epistasis.org (https://github.com/EpistasisLab/scikit-mdr or https://ritchielab.org/software/mdr-downloads-1, accessed on 8 October 2023). The MDR method can identify associations in studies with small sample numbers and low penetrance SNPs. This method assumes a non-parametric and model-less machine learning approach designed to detect the characterization of non-additive gene–gene interactions in the absence of statistically detectable independent effects. To assess the predictive accuracy, ten-fold cross-validation was applied. The best model that maximized the accuracy of the balance of evidence (TBA) or the consistency of cross-validation (CVC), as well as the sensitivity and specificity values, was selected. All results were considered statistically significant at *p* < 0.05. Calculation of the network of combined MDR attributes or calculation of Cartesian products was carried out by selecting the best threshold (>1) to identify the entropy between the genes.

### 4.5. Evaluation of Cognitive Function in Patients with Epilepsy

Neuropsychological outcomes in 42 of 198 patients with TLE were assessed using a neuropsychological assessment battery following resective epilepsy surgery. The sociodemographic information of the patients is summarized in Appendix A. The neuropsychological battery included standardized tests described in Appendix A, classified into the following domains: orientation, attention, expressive language, impressionistic language, written language, gnosis, praxis, memory, and executive functions. The evaluated domains represented key cognitive functions and were scored according to standard norms. The results from each test were classified into general indices: no cognitive impairment, mild, moderate, and severe impairment. Logistic regression was used for statistical analysis, adjusting for age, sex, and educational level.

## 5. Conclusions

Collectively, our data show that the *APOE ε4* allele can interact with genetic variants located in the *MAPT* and *GSK3B* loci, suggesting an important role of tau hyperphosphorylation in conferring genetic susceptibility in AD and refractory TLE (Figure 3). It is likely that specific interactions can be defined and used to discriminate each pathology. Therefore, future studies must be conducted to validate the genetic interactions in a large cohort of patients, including correlations with functional assays. If replicated, these findings may also support previous proposals to develop tau phosphorylation pharmacotherapy strategies to treat refractory temporal lobe epilepsy.

## Figures and Tables

**Figure 1 ijms-25-10228-f001:**
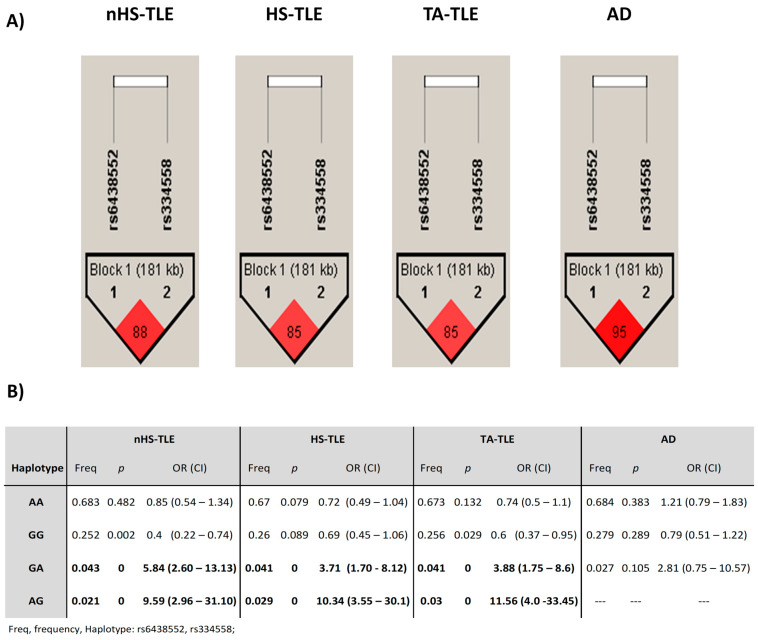
Linkage disequilibrium (LD) of GSK3B SNPs in epilepsy and Alzheimer’s disease. (**A**) D’ values are shown within cells. The standard LD color scheme was used, with white to red colors representing increasing LD strength. Significant values, represented by haplotype blocks, are observed between variants rs334558 and rs6438552 located in the GSK3B gene. (**B**) Significant haplotypes for each study group. Values are in bold to emphasize their significance.

**Figure 2 ijms-25-10228-f002:**
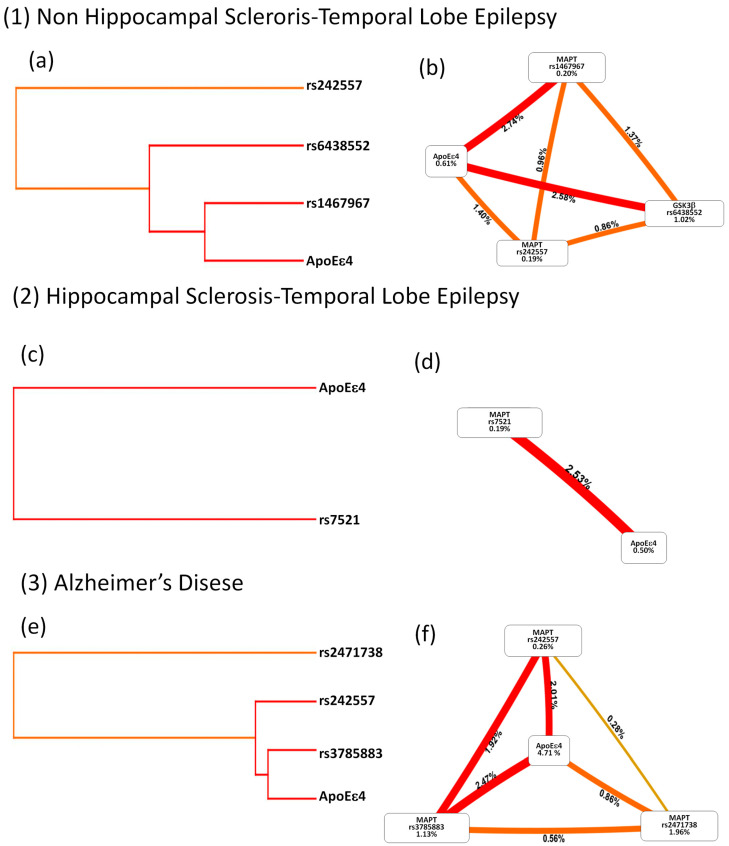
Interaction network diagram in the genetic variants and patients according to the MDR analysis. Patients with nHS-TLE (1), HS-TLE (2) and AD (3) compared to their controls. (**a**,**c**,**e**), Dendrogram interaction plots, generated by hierarchical cluster analysis, illustrate presence, strength and type of epistatic effects. The color of the line indicates the type of interaction. Red and orange indicate a synergistic relationship (i.e., epistasis). Green and blue suggest redundancy or linkage. (**b**,**d**,**f**) Circular graphs; the percentage at the bottom of each variable represents its entropy, and the percentage on each line represents the interaction of the percentage of entropy between two variables.

**Figure 3 ijms-25-10228-f003:**
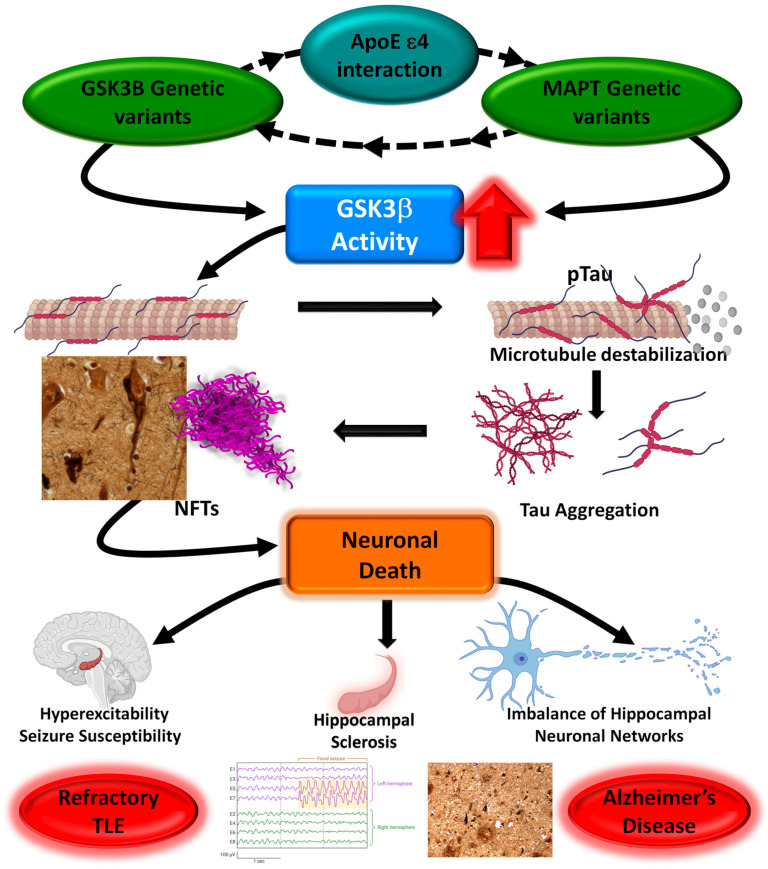
Schematic of tau-GSK3b interaction in Alzheimer’s disease and epilepsy. Overactivation of GSK3b promotes tau phosphorylation and aggregation into neurofibrillary tangles (NFTs), which induce neuronal death and imbalance hippocampal neuronal networks, promoting hyperexcitability and seizure development.

**Table 1 ijms-25-10228-t001:** Demographic data of the population.

	nHS-TLE	HS-TLE	TA-TLE	Controls	AD	AD Controls
	*n* = 49	*n* = 79	*n* = 70	*n* = 245	*n* = 100	*n* = 106
Age (years)	36.8 ± 9.87	38.2 ± 10.2	37.33 ± 12.22	38.8 ± 20.08	75.37 ± 10.35	73.53 ± 10.22
Mean ± SD						
min–max	22–66	19–68	17–69	18–91	32–100	43–100
Median (p25–p75)	36 (29–41)	37 (30–45)	36.5 (28–46)	30 (24–47)	75.5 (68.5–83)	73.5 (68–83)
*p*-value	0.101	**0.011**	0.167		0.178	
Sex N (%)						
Males	23 (47%)	42 (53.2%)	36 (51.4%)	103 (42%)	32 (32%)	39 (36.8%)
Females	26 (53%)	37 (46.8%)	34 (48.6%)	142 (58%)	68 (68%)	67 (63.2%)
x^2^	0.4	2.99	1.94		0.52	
*p*-value	0.52	0.08	0.16		0.47	

nHS-TLE; non-Hippocampal Sclerosis-Temporal Lobe Epilepsy. HS-TLE; Hippocampal Sclerosis-Temporal Lobe Epilepsy. TA-TLE; Tumor-Associated-Temporal Lobe Epilepsy. AD; Alzheimer’s disease. *p* < 0.05; The *p*-value, group of patients vs. control for each case Mann-Whitney U. Values are in bold to emphasize their significance.

**Table 2 ijms-25-10228-t002:** Allele and genotypic frequencies were obtained for each polymorphism in the group of cases and controls.

	Polymorphisms Alleles/Genotypes	nHS-TLE	HS-TLE	TA-TLE	Controls	AD	AD Controls
Gene		*n* = 49	*n* = 79	*n* = 70	*n* = 245	*n* = 100	*n* = 106
** *MAPT* **	**rs242557**	** *n* ** **(%)**	** *n* ** **(%)**	** *n* ** **(%)**	** *n* ** **(%)**	** *n* ** **(%)**	** *n* ** **(%)**
	A/G	32 (33)/66 (67)	63 (39)/96 (61)	51 (36)/90 (64)	182 (37)/308 (63)	74 (37)/126 (63)	71 (33)/142 (67)
	AA/GA/GG	5 (10)/22 (45)/22 (45)	10 (12.7)/42 (53.2)/27 (34.2)	6 (8.6)/38 (54.3)/26 (37.1)	31 (13)/120 (49)/94 (38)	13 (13)/48 (48)/39 (39)	11 (10.4)/48 (45.3)/47 (44.3)
	*p*	0.68	0.78	0.57		0.69	
	EHW *p*	1	0.36	0.19	0.5	0.83	1
	**rs1467967**						
	A/G	67 (68)/31 (32)	101 (64)/57 (36)	88 (63)/52 (37)	312 (64)/178 (36)	123 (62)/77 (38)	144 (68)/68 (32)
	AA/AG/GG	24 (49)/19 (39)/6 (12)	33 (42)/35 (44)/11 (14)	29 (41)/30 (43)/11 (16)	103 (42)/106 (43)/36 (15)	41 (41)/41 (41)/18 (18)	52 (49.1)/40 (37.7)/14 (13.2)
	*p*	0.66	0.98	0.98		0.44	
	EHW *p*	0.51	0.81	0.61	0.33	0.2	0.18
	**rs2471738**						
	C/T	72 (72)/27 (28)	107 (67)/52 (33)	97 (69)/44 (31	322 (66)/169 (34)	148 (74)/53 (26)	136 (64)/77 (36)
	CC/CT/TT	25 (51)/21 (42.9)/3 (6.1)	36 (45.6)/34 (43)/9 (11.4)	35 (50.0)/26 (37.1)/9 (12.9)	105 (42.9)/111 (45.3)/29 (11.8)	57 (57)/33 (33)/10 (10)	14 (13.2)/49 (46.2)/43 (40.6)
	*p*	0.38	0.91	0.47		0.06	
	EHW *p*	0.73	0.8	0.27	1	0.13	1
	**rs7521**						
	A/G	30 (31)/68 (69)	49 (31)/109 (69)	44 (31)/96 (69)	145 (30)/345 (70)	56 (28)/144 (72)	61 (29)/151(71)
	AA/GA/GG	5 (10.2)/20 (40.8)/24 (49.0)	4 (5.1)/41 (51.9)/34 (43)	6 (8.6)/32 (45.7)/32 (45.7)	16 (6.5)/113 (46.1)/116 (47.3)	12 (12)/32 (32)/56 (56)	8 (7.5)/45 (42.5)/53 (50.0)
	*p*	0.59	0.65	0.836		0.23	
	EHW *p*	0.74	0.11	0.78	0.12	**0.046**	0.82
	**rs3785883**						
	A/G	16 (16)/82 (84)	22 (14)/136 (86)	20 (14)/120 (86)	81 (17)/409 (83)	30 (15)/170 (85)	46 (22)/166 (78)
	AA/GA/GG	2 (4.1)/12 (24.5)/35 (71.4)	2 (2.5)/18 (22.8)/59 (74.7)	2 (2.9)/16 (22.9)/52 (74.3)	7 (2.9)/67 (27.3)/171 (69.8)	2 (2)/26 (26)/72 (72)	4 (3.8)/38 (35.8)/64 (60.4)
	*p*	0.84	0.7	0.75		0.2	
	EHW *p*	0.59	0.64	0.62	0.82	1	0.78
** *GSK3B* **	**rs334558**						
	A/ G	77 (79)/21 (21)	111 (70)/47 (30)	100 (71)/40 (29)	350 (71)/140 (29)	149 (74)/51 (26)	144 (0.68)/68 (0.32)
	AA/AG/GG	31 (63.3)/15 (30.6)/3 (6.1)	38 (48.1)/35 (44.3)/6 (7.6)	39 (55.7)/22 (31.4)/9 (12.9)	126 (51.4)/98 (40)/21 (8.6)	59 (59)/31 (31)/	52 (49.1)/40 (37.7)/14 (13.2)
	p	0.31	0.79	0.32		0.35	
	EHW *p*	0.67	0.79	0.075	0.76	0.068	0.18
	**rs6438552**						
	A/G	72 (73)/26 (27)	110 (71)/46 (29)	101 (72)/39 (28)	342 (70)/148 (30)	141 (70)/59 (30)	145 (68)/67 (32)
	AA/AG/GG	24 (49)/24 (49)/1 (2)	40 (50.6)/31 (39.2)/8 (10.1)	35 (50.0)/31 (44.3)/4 (5.7)	120 (49)/102 (41.6)/23 (9.3)	52 (52)/37 (37)/11 (11)	53 (50.0)/39 (36.8)/14 (13.2)
	*p*	0.2	0.93	0.62		0.88	
	EHW *p*	0.14	0.58	0.55	0.88	0.33	0.12
** *APOE* **	**APOE**						
	E2/E2	1 (2)	2 (2.5)	0	3 (1.2)	0 (0)	1 (1)
	E2/E3	0 (0)	1 (1.3)	0	27 (11)	7 (7)	44 (41.9)
	E3/E3	34 (69.4)	67 (84.8)	56 (80)	170 (69.4)	57 (57)	45 (42.9)
	E3/E4	14 (28.6)	9 (11.4)	14 (20)	43 (17.6)	28 (28)	14 (13.3)
	E4/E4	0	0	0	2 (0.8)	8 (8)	0 (0)
	E2/E4	0	0	0	0	0 (0)	1 (1)
	*p*	0.073	**0.024**	**0.038**		**0.0001**	

Non-Hippocampal Sclerosis-Temporal lobe epilepsy (nHS-TLE), Hippocampal Sclerosis-Temporal Lobe Epilepsy (HS-TLE), Tumor-Associated Temporal Lobe Epilepsy (TA-TLE), Alzheimer’s disease (AD), and a control group. Statistical analyses were performed using SPSS software. A x^2^ test was performed for genotype frequencies, and *p* values < 0.05 are highlighted in bold. EHW *p*: The *p* of the Hardy–Weinberg equilibrium was calculated, genes that do not comply with the equilibrium law are highlighted in bold, values of *p* < 0.05 are considered significant *p* < 0.05; The *p*-value, group of patients vs. control for each case. Values are in bold to emphasize their significance.

**Table 3 ijms-25-10228-t003:** The *p* and OR of the genotypes were calculated for each group of cases and controls.

		**nHS-TLE**	**HS-TLE**	**TA-TLE**	**AD**
		** *n* ** **= 49**	** *n* ** **= 79**	** *n* ** **= 70**	** *n* ** **= 100**
**Gene**	** *Polymorphisms* **				
** *MAPT* **	**rs242557**				
	**MODELS**	**[OR(CI), *p*]**	**[OR(CI), *p*]**	**[OR(CI), *p*]**	**[OR(CI), *p*]**
	**D**				
	“A/G+A/A” vs. G/G	0.507 (0.244–1.054), 0.069	1.242 (0.696–2.217), 0.464	1.171 (0.624–2.198), 0.624	0.806 (0.419–1.548), 0.516
	“A/G+A/A” vs. G/G * APOE	4.22 (0.845–21.068), 0.079	0.568 (0.118–2.722), 0.479	0.488 (0.124–1.930), 0.307	**4.682 (1.104–19.852), 0.036**
	**R**				
	A/A vs. “A/G+G/G”	0.358 (0.081–1.580), 0.175	0.862 (0.382–1.084), 0.721	0.557 (0.203–1.530), 0.256	1.487 (0.576–3.840), 0.413
	A/A vs. “A/G+G/G” * APOE	**11.366 (1.147–112.648), 0.038**	2.128 (0.168–26.78), 0.559	1.737 (0.134–22.574), 0.673	0.866 (0.069–10.930), 0.912
	**H**				
	A/G vs. “A/A+G/G”	0.729 (0.349–1.522), 0.401	1.305 (0.753–2.26), 0.343	1.464 (0.803–2.669), 0.214	0.661 (0.340–1.287), 0.223
	A/G vs. “A/A+G/G” * APOE	1.359 (0.327–5.636), 0.673	0.452 (0.096–2.126), 0.315	0.396 (0.102–1.545), 0.182	**4.862 (1.171–20.187), 0.029**
	**rs1467967**				
	**D**				
	“A/G+G/G” vs. A/A	0.671 (0.326–1.380), 0.278	0.971 (0.557–1.694), 0.919	1.007 (0.550–1.844), 0.982	1.244 (0.647–2.391), 0.513
	“A/G+G/G” vs. A/A * APOE	1.640 (0.399–6.733), 0.493	1.019 (0.217–4.797), 0.981	1.105 (0.284- 4.302), 0.886	1.559 (0.388–6.261), 0.531
	**R**				
	G/G vs. “A/G+A/A”	0.150 (0.20–1.138), 0.066	0.880 (0.405–1.910), 0.747	1.131 (0.515–2.481), 0.756	1.441 (605–3.432), 0.41
	G/G vs. “A/G+A/A” * APOE	**47.717 (3.727–610.877), 0.003**	1.799 (0.153–21.088), 0.64	0.759 (0.067–8.531), 0.823	2.319 (0.213–25.279), 0.49
	**H**				
	A/G vs. “A/A+G/G”	1.152 (0.559–2.376), 0.701	1.1 (0.633–1.911), 0.735	0.938 (0.513–1.716), 0.836	1.03 (0.518–1.980), 0.969
	A/G vs. “A/A+G/G” * APOE	0.257 (0.053–1.255), 0.093	0.785 (0.167–3.693), 0.759	1.237 (0.321–4.763), 0.757	1.173 (0.286–4.804), 0.824
	**rs2471738**				
	**D**				
	“T/C+T/T” vs. C/C	0.647 (0.314–1.33), 0.238	0.711 (0.410–1.235), 0.226	0.684 (0.376–1.243), 0.213	**0.458 (0.237- 0.888), 0.021**
	“T/C+T/T” vs. C/C * APOE	1.71 (0.420–6.966), 0.454	5.375 (0.917–31.522), 0.062	1.562 (0.406–6.010), 0.517	2.786 (0.658–11.791), 0.164
	**R**				
	T/T vs. “T/C+C/C”	0.437 (0.098–1.940), 0.276	0.659 (0.256–1.694), 0.387	0.855 (0.330–2.215), 0.747	0.943 (0.340–2.616), 0.91
	T/T vs. “T/C+C/C” * APOE	1.356 (0.092–20.028), 0.825	5.748 (0.840–39.331), 0.075	2.411 (0.380–15.319), 0.351	0.374 (0.050–2.813), 0.339
	**H**				
	T/C vs. “T/T+C/C”	0.840 (0.406–1.739), 0.638	0.832 (0.478–1.447), 0.515	0.727 (0.396–1.334), 0.303	**0.457 (0.232–0.899), 0.023**
	T/C vs. “T/T+C/C” * APOE	1.587 (0.381–6.606), 0.525	1.711 (0.361–8.111), 0.499	0.952 (0.223–4.061), 0.947	**6.4 (1.063–38.798), 0.043**
	**rs7521**				
	**D**				
	“A/G+A/A” vs. G/G	1.094 (0.531–2.253), 0.807	1.639 (0.937–2.868), 0.083	1.278 (0.702–2.327), 0.423	0.687 (0.356–1.323), 0.262
	“A/G+A/A” vs. G/G * APOE	0.505 (0.123–2.080), 0.344	**0.093 (0.016–0.556), 0.009**	0.437 (0.112–1.695), 0.231	1.129 (0.281–4.543), 0.864
	**R**				
	A/A vs. “A/G+G/G”	1.928 (0.582–6.388), 0.283	0.660 (0.182–2.393), 0.573	1.845 (0.660–5.154), 0.243	1.036 (0.339–3.165), 0.95
	A/A vs. “A/G+G/G” * APOE	0.916 (0.058–14.405), 0.95	4.093 (0.239–70.014), 0.331		
	**H**				
	A/G vs. “A/A+G/G”	0.888 (0.427–1.849), 0.751	**1.77 (1.0018–3.076), 0.043**	1.060 (0.582–1.93), 0.849	0.654 (0.328–1.303), 0.227
	A/G vs. “A/A+G/G” * APOE	0.555 (0.133–2.312), 0.419	**0.046 (0.005–0.428) *p* = 0.007**	0.628 (0.163–2.422), 0.499	0.532 (0.128–2.217), 0. 386
	**rs3785883**				
	**D**				
	“A/G+A/A” vs. G/G	0.746 (0.329–1.69), 0.482	0.680 (0.363–1.273), 0.228	0.722 (0.367–1.420), 0.345	0.92 (0.469–1.805), 0.809
	“A/G+A/A” vs. G/G * APOE	2.338 (0.507–10.78), 0.276	2.657 (0.493–14.316), 0.255	1.919 (0.421–8.75), 0.4	**0.155 (0.034–0.710), 0.016**
	**R**				
	A/A vs. “A/G+G/G”	1.862 (363–9.553), 0.456	0.397 (0.040–3.332), 0.395	1.131 (0.224–5.702), 0.882	0.897 (0.144–5.587), 0.907
	A/A vs. “A/G+G/G” * APOE				
	**H**				
	A/G vs. “A/A+G/G”	0.624 (0.257–1.517), 0.298	0.743 (0.391–1.410), 0.363	0.685 (0.336–1.398), 0.299	0.933 (0.467–1.862), 0.844
	A/G vs. “A/A+G/G” * APOE	2.796 (0.583–13.411), 0.199	1.360 (0.215–8.581), 0.744	2.026 (0.436–9.413), 0.368	**0.204 (0.045–0.930), 0.04**
** *GSK3B* **	**rs334558**				
	**D**				
	“A/G+G/G” vs. A/A	**0.477 (0.221–1.029), 0.059**	1.187 (0.684–2.060), 0.542	0.890 (0.499–1.618), 0.702	0.701 (0.366–1.344), 0.285
	“A/G+G/G” vs. A/A * APOE	2.53 (0.604–10.601), 0.204	0.807 (0.171–3.801), 0.786	0.733 (0.183–2.931), 0.66	0.689 (0.170–2.794), 0.602
	**R**				
	G/G vs. “A/G+A/A”	0.706 (0.153–3.252), 0.655	0.866 (0.302–2.483), 0.789	2.254 (0.926–5.485), 0.073	0.603 (0.215–1.695), 0.338
	G/G vs. “A/G+A/A” * APOE	1.079 (0.069–16.927), 0.957	1.449 (0.112–18.729), 0.777		2.677 (0.218–32.822), 0.441
	**H**				
	A/G vs. “A/A+G/G”	0.499 (0.222–1.120), 0.092	1.238 (0.713–2.149), 0.448	0.616 (0.326–1.167), 0.137	0.853 (0.429–1.694), 0.649
	A/G vs. “A/A+G/G” *	2.643 (0.608–11.483), 0.195	0.705 (0.140–3.557), 0.673	1.563 (0.382–6.402), 0.535	0.473 (0.114–1.968), 0.303
	**rs6438552**				
	**D**				
	“A/G+G/G” vs. A/A	0.582 (0.276–1.224), 0.153	1.054 (0.609–1.824), 0.852	1.226 (0.673–2.234), 0.506	0.961 (0.504–1.830), 0.903
	“A/G+G/G” vs. A/A * APOE	**9.549 (1.613–56.539), 0.013**	0.443 (0.089–2.210), 0.321	0.285 (0.067–1.207), 0.088	0.723 (0.179–2.913), 0.648
	**R**				
	G/G vs. “A/G+A/A”	0.337 (0.043–2.635), 0.3	1.442 (0.587–3.542), 0.425	0.876 (0.278–2.756), 0.821	0.724 (0.269–1.948), 0.523
	G/G vs. “A/G+A/A” * APOE				2.233 (0.185–26.897), 0.527
	**H**				
	A/G vs. “A/A+G/G”	0.714 (0.335–1.519), 0.381	0.924 (0.529–1.615), 0.782	1.27 (0.699–2.307), 0.433	1.12 (0.569–2.203), 0.743
	A/G vs. “A/A+G/G” * APOE	**13.523 (2.265–80.736), 0.004**	0.871 (0.173–4.382), 0.867	0.478 (0.112–2.028), 0.317	0.531 (0.130–2.163), 0.377
** *APOE* **	**APOE**				
	MODELS				
	ApoE4 carriers vs. ApoE4 non-carriers E4	1.8 (0.892–1.013)	0.588 (0.271–1.271)	1.138 (0.580–2.233)	**3.458 (1.734–6.897)**
	*p*	0.1	0.177	0.707	**0.0001**
**INTERACTION**	**nHS-TLE**				
	** *MAPT_GSK3B_APOE* **			**[OR(CI), *p*]**	
	rs1467967 (G/G vs. “A/G+A/A”)_ rs6438552 (“A/G+G/G” vs. A/A)_APOE			**23.401 (2.496–219.402), 0.006**	

Logistic regression was performed by adjusting for age and sex using the SPSS statistical program. Values that are *p* < 0.05 are highlighted in bold, considering them significant. The diseases (nHS-TLE, HS-TLE, TA-TLE, and AD) are always compared with their control group. Association under three genetic models: dominant (D), recessive (R), and heterozygote (H). *ApoE* was calculated *p* and OR for each of the diseases (nHS-TLE, HS-TLE, TA-TLE, and AD) adjusted for age and sex. Values of *p* < 0.05 are indicated in bold, considering them as statistically significant. nHS-TLE; non-Hippocampal Sclerosis-Temporal Lobe Epilepsy. HS-TLE; Hippocampal Sclerosis-Temporal Lobe Epilepsy. TA-TLE; Tumor Associated-Temporal Lobe Epilepsy. AD; Alzheimer’s disease. *p* < 0.05; *p*-value, group of patients vs. control for each case. The symbol * refers to the additive effect that *APOE ε4* has on the inheritance. Values are in bold to emphasize their significance.

**Table 4 ijms-25-10228-t004:** Gene–Gene Interaction.

Model	Accuracy	Sensitivity	Specificity	Consistency	OR (95%CI)	*p*
**Non-Hippocampal Sclerosis-Temporal Lobe Epilepsy**						
*APOE*	0.686	0.354	0.752	7/10	1.663 (0.835–3.312)	0.145
*GSK3B*_rs6438552, *APOE* (G*ε4 carriers)	0.556	0.696	0.528	9/10	2.566 (1.282–5.137)	0.006
*MAPT2*_rs1467967, *GSK3B*_rs6438552, *APOE* (GG*ε4)	0.652	0.696	0.643	10/10	4.136 (2.057–8.316)	<0.001
*MAPT1*_rs242557, *MAPT2*_rs1467967, *GSK3B*_rs6438552, *APOE* (AGG*ε4)	0.632	0.809	0.597	10/10	**6.291 (2.831–13.97)**	**<0.001**
**Hippocampal Sclerosis-Temporal Lobe Epilepsy**						
*APOE*	0.413	0.783	0.294	7/10	1.505 (0.798–2.839)	0.204
*MAPT4*_rs7521, *APOE* (A*ε4)	0.588	0.601	0.584	10/10	**2.108 (1.222–3.635)**	**0.007**
**Patients with Alzheimer’s Disease**						
*APOE*	0.616	0.36	0.858	10/10	3.412 (1.663–7.003)	<0.001
*MAPT3*_rs2471738, *APOE* (T*ε4)	0.632	0.59	0.671	7/10	2.933 (1.611–5.339)	<0.001
*MAPT3*_rs2471738, *MAPT5*_rs3785883, *APOE* (TA*ε4)	0.658	0.628	0.686	6/10	3.695 (2.011–6.787)	<0.001
*MAPT1*_rs242557, *MAPT3*_rs2471738, *MAPT5*_rs3785883, *APOE* (ATA*ε4)	0.683	0.547	0.812	10/10	**5.221 (2.697–10.108)**	**<0.001**

OR, odds ratio; CI, confidence interval. * represents the interaction between *APOE ε*4 with corresponding allele or genotype. Values are in bold to emphasize their significance.

## Data Availability

The original contributions presented in the study are included in the article/Appendix A, further inquiries can be directed to the corresponding author/s.

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
