# Peer review of "Synergistic Effect between the APOE ε4 Allele with Genetic Variants of GSK3B and MAPT: Differential Profile between Refractory Epilepsy and Alzheimer Disease"

_ijms, 2024, doi:10.3390/ijms251810228_

Round 1

Reviewer 1 Report

Comments and Suggestions for Authors

There is a confusing organization in the description of this research. Generally, the research articles are divided into introduction, materials and methods, results and discussion. In this manuscript, this order is different and make difficult to read the study design and its findings. Moreover, there are some typos and writing errors.

Materials and methods section:

This section must be improved. The authors should clarify the inclusion and exclusion criteria used for this study design. How old are the recruited patients with AD and epilepsy? How the control group was recruited? Why the control group was divided in two and matched separately with the AD- and epilepsy groups? How was assessed the cognitive function in patients with epilepsy? Which neuropsychological assessment battery?

Results section: 

This section can be improved. The authors should only describe the results obtained from their analysis. Their discussion should be included in the appropriate section. (see, for example, the sentence "the apolipoprotein E gene..."in line 101-103 is non pertinent in this section and already cited in previous paragraph). 

Discussion section:

Strenghs and limitations of this study, if any, should be discussed.

Author Response

We thank all reviewers for the time and effort in reviewing our manuscript. The criticisms are very thoughtful and constructive. We believe modifying our manuscript to address these criticisms has improved our manuscript.

Point-by-point responses to reviewer comments are in blue italics.

REVIEWER 1

Comments and Suggestions for Authors:

There is a confusing organization in the description of this research. Generally, the research articles are divided into introduction, materials and methods, results and discussion. In this manuscript, this order is different and make difficult to read the study design and its findings. Moreover, there are some typos and writing errors.

We thank Reviewer 1 for the thoughtful and constructive criticisms of our manuscript. We have corrected several typos and writing errors. The changes are highlighted in yellow, as well as identified in track changes. Regarding the organization of the paper, we agreed that following the traditional order for the sections improve the understanding of our research instead of using the suggested format template from the journal. The materials and methods sections were moved right after the introduction.

Materials and methods section:

This section must be improved. The authors should clarify the inclusion and exclusion criteria used for this study design. How old are the recruited patients with AD and epilepsy? How was the control group recruited? Why was the control group divided in two and matched separately with the AD- and epilepsy groups?

We thank the reviewer for the suggestion. We have now added further information to address the concerns about the inclusion and exclusion criteria, as well as additional information of patients and controls. Patients with Alzheimer's disease have an age range of ≥60-90 years. The age range of patients with epilepsy varies from 18-60 years. The controls were recruited in an open registration published at the INNN and HGM, following the inclusion and exclusion criteria described in this section. The main reason of including a control group for LOAD patients and another for TLE patients was for match them by age, a brief explanation was added in the control group’s description. The changes are in the following section.

Subjects.

We performed a genetic case-control study on unrelated individuals from Mexico City who knew how to read and write, and provided informed consent to participate in this study and whose families have born and lived in Mexico for at least three generations.

  1. Patients: Late-onset Alzheimer’s Disease (LOAD) patients and Temporal Lobe Epilepsy (TLE) patients were analyzed in the present genetic study.

Late-onset Alzheimer’s disease patients (age-onset ≥60 years, n=100) were recruited from the General Hospital of Mexico (HGM), Hospital Angeles Mocel, and the National Institute of Neurology and Neurosurgery of Mexico (INNN), all of them located in Mexico City. The inclusion criteria were as follows: (1) individuals over 60 years of age (2) who had been clinically diagnosed with AD by a group of experts, using the criteria of the National Institute of Neurological and Communicative Diseases and Stroke/Alzheimer's Disease and Related Disorders Association (NINCDS-ADRDA)[61]. Subjects with any other type of neurodegenerative or neuroinfectious disease and patients with neoplasia were excluded from this study.

Patients clinically diagnosed with refractory Temporal Lobe Epilepsy (18-60 years, n=198) and that underwent epilepsy resective surgery were recruited from the Epilepsy Priority Program 'PPE' of the INNN and the HGM hospital. The inclusion criteria of clinically diagnosed individuals with refractory epilepsy were as follows: (1) defined as having received at least two drug regimens at appropriate therapeutic doses for at least 6 months, (2) with neurological monitoring, and (3)having seizures at a frequency of at least 3 per month [62]. The TLE cases were stratified according to their pathological findings in Hippocampal Sclerosis-TLE patients (HS-TLE, n=79), non-Hippocampal Sclerosis-TLE patients (nHS-TLE, n=49) and Tumor Associated-TLE (TA-TLE, n=70). Subjects with any other type of additional neurodegenerative disease were excluded from the study.

  1. Healthy individuals were recruited as controls, stratified by age in two groups as follows: Controls for LOAD patients and Controls for TLE patients.

Controls for the LOAD patients (n=106) were recruited at the INNN and the HGM hospitals following the next inclusion criteria: (1) subjects older than 60 years, (2) clinically diagnosed as non-demented according to the NINCDS-ADRD criteria. Subjects diagnosed with any neurodegenerative disease or with a family genetic history of a neurodegenerative disease were excluded from the study.

Controls for the TLE patients (n=245) were recruited at the INNN and the HGM hospitals following the next inclusion criteria: (1) subjects around 18 and 60 years old (2) clinically diagnosed as healthy. Subjects diagnosed with any neurological disease or with family genetic history of a neurological disease were excluded from the study.

How was assessed the cognitive function in patients with epilepsy? Which neuropsychological assessment battery?

We thank Reviewer 1 to point out a lack of information in the description for the neuropsychological tests applied in part of our TLE patients. The battery includes tests that evaluate important cognitive functions and are mentioned in Table 4S (i.e., the Wechsler Memory Scale, the Montreal Cognitive Assessment and the Barcelona test). The new proposal description for the battery is included in the Methods section as follows:

Evaluation of cognitive function in patients with epilepsy.

Neuropsychological outcomes in 42 of the 198 TLE patients were performed with a neuropsychological assessment battery following the resective epilepsy surgery. The sociodemographic information from the patients is summarized in Table 3S. The neuropsychological battery included standardized tests described in Table 4S, classified into the following domains: orientation, attention, expressive language, impressionistic language, written language, gnosis, praxis, memory and executive functions). The evaluated domains represent key cognitive functions and were scored according to standard norms. Results from each test were classified in general indices as no cognitive impairment, mild, moderate, and severe impairment. A logistic regression was used for statistical analysis, adjusting for age, sex, and educational level.

Results section:

This section can be improved. The authors should only describe the results obtained from their analysis. Their discussion should be included in the appropriate section. (see, for example, the sentence "the apolipoprotein E gene..."in line 101-103 is non pertinent in this section and already cited in previous paragraph).

We thank Reviewer 1 for pointing this out. After checking the results section, we eliminate the next paragraphs as they are part from the discussion.

In page 5, line 190 (previously line 101-103)…

The apolipoprotein E gene (APOE) is the strongest genetic risk factor for AD. Presence of APOE ε4 allele has been associated with about 2-fold increased risk of developing AD in Hispanic populations [22] As expected,

In page 9, line 1-2

The APOE ε4 allele has been associated with further risk of developing Alzheimer-type neuropathological changes in patients with epilepsy [21, 23]. Regarding,

Discussion section:

Strengths and limitations of this study, if any, should be discussed

Thanks for your suggestion. We agreed that a paragraph in the discussion section was necessary to line the strengths limitations of our pilot study.

The identification of an interaction between the e4 allele of APOE and the genetic variants of GSK3B and MAPT is one of the main strengths of this study. This interaction also allowed us to observe a specific genetic profile between the different epilepsy groups and AD patients. This finding may, in the future, open new lines of research that will help to identify specific therapeutics for each pathology. At the same time, it contributes to studying a population with low representativeness in analyzing possible genetic factors associated with the risk of developing epilepsy and AD in the Mexican population. The main limitation of this work was the sample size, as well as the extension of the analysis of cognitive tests that allowed the establishment of a close correlation between the findings with cognitive impairment.

Reviewer 2 Report

Comments and Suggestions for Authors

In Toral-Rios et al., the authors aimed to study whether the presence of single nucleotide polymorphisms (SNPs) in genes of the microtubule-associated protein tau (MAPT), the kinase GSKB, and two heat shock proteins involved in tau aggregation (HSPs) are associated with the risk of developing AD and drug-resistant TLE in a cohort of Mexican population. For this, the authors investigated whether APOE ɛ4 allele and genetic variants located in the MAPT and GSK3B gene are associated with the risk of developing AD and drug-resistant TLE in the same cohort. The authors state that the major genetic risk factor in late-onset AD is the ε4 allele of Apolipoprotein E, whereas, its presence is also associated with the development of refractory epilepsy. The data of this manuscript show that APOE e4 allele can interact with genetic variants located in MAPT and GSK3B loci, suggesting an important role of tau hyperphosphorylation in conferring genetic susceptibility in AD and refractory TLE. The study is well designed and presented. This study is of some importance in the field, as these findings may be used to advance tau phosphorylation pharmacotherapy strategies to treat refractory temporal lobe epilepsy.

While the review is interesting, I have a concern.

    Adding a pictorial diagram representing how developing AD and drug-resistant TLE are intertwined together would make for a good read.

Author Response

We thank all reviewers for the time and effort in reviewing our manuscript. The criticisms are very thoughtful and constructive. We believe modifying our manuscript to address these criticisms has improved our manuscript.

Point-by-point responses to reviewer comments are in blue italics.

REVIEWER 2

In Toral-Rios et al., the authors aimed to study whether the presence of single nucleotide polymorphisms (SNPs) in genes of the microtubule-associated protein tau (MAPT), the kinase GSKB, and two heat shock proteins involved in tau aggregation (HSPs) are associated with the risk of developing AD and drug-resistant TLE in a cohort of Mexican population. For this, the authors investigated whether APOE ɛ4 allele and genetic variants located in the MAPT and GSK3B gene are associated with the risk of developing AD and drug-resistant TLE in the same cohort. The authors state that the major genetic risk factor in late-onset AD is the ε4 allele of Apolipoprotein E, whereas, its presence is also associated with the development of refractory epilepsy. The data of this manuscript show that APOE e4 allele can interact with genetic variants located in MAPT and GSK3B loci, suggesting an important role of tau hyperphosphorylation in conferring genetic susceptibility in AD and refractory TLE. The study is well designed and presented. This study is of some importance in the field, as these findings may be used to advance tau phosphorylation pharmacotherapy strategies to treat refractory temporal lobe epilepsy.

While the review is interesting, I have a concern.
Adding a pictorial diagram representing how developing AD and drug-resistant TLE are intertwined together would make for a good read.

We thank Reviewer 2 for the positive comments about our manuscript and a thorough review. In addition, thanks for suggesting a diagram to better understand how AD and drug-resistant TLE shared genetic risk factors (Figure 3).

Round 2

Reviewer 1 Report

Comments and Suggestions for Authors

The manuscript was clearly improved, especially in the form and quality of presentation. Suggestions have been heard and the necessary changes made, making the study research clearer and better discussed.